# NOVEL DOMAIN EXTRAPOLATION WITH LARGE LANGUAGE MODELS

## ABSTRACT

We study Domain Generalization (DG), which evaluates models' ability to generalize to unseen test domains. Various augmentation strategies, such as domain augmentation, have been proposed to mitigate this issue. However, many of these methods largely rely on interpolating existing domains and frequently face difficulties in creating truly "novel" domains.

We introduce a novel approach to domain extrapolation that leverages the extensive knowledge encapsulated within large language models (LLMs) to synthesize entirely new domains. Starting with the class of interest, we query the LLMs to extract relevant knowledge for these novel domains. We then bridge the gap between the text-centric knowledge derived from LLMs and the pixel input space of the model using text-to-image generation techniques. By augmenting the training set of domain generalization datasets with high-fidelity, photo-realistic images of these new domains, we achieve significant improvements over all existing methods. This is demonstrated in both single and multi-domain generalization across various benchmarks.

Our empirical findings support our argument that the knowledge from the LLMs and a realization that can bridge the text-driven knowledge and the pixel input space is adequate to learn a generalized model for any task. To illustrate, we put forth a much more difficult setting termed, **data-free domain generalization**, that aims to learn a generalized model in the absence of any collected data. Surprisingly, our proposed method exhibits commendable performance in this setting, even surpassing the supervised setting by approximately 1-2% on datasets such as VLCS.

## 1 INTRODUCTION

Domain generalization (DG) (Blanchard et al., 2011) studies objective of learning a model from multiple source domains that can generalize to unseen testing domains. While the idea behind DG is to utilize multiple source domains to help models recognize similarities and differences, thus aiding in generalization to new test domains, a significant challenge arises. The availability of these source domains often becomes a limiting factor, hindering the success of current DG approaches in more challenging scenarios Qiao et al. (2020); Wang et al. (2021); Xu et al. (2020); Wang et al. (2022). Moreover, the unavailability of multiple source domains presents a pragmatic challenge as it's labor-intensive and costly to collect, not just, data but data in diverse domains with annotations. In addition, the collection is sometimes even impossible in critical areas such as healthcare or extreme conditions (e.g. deep sea or space). This motivates Single Domain Generalization Fan et al. (2021); Wang et al. (2022) which aims to generalize to unseen testing domains with a single training domain.

Motivated by these challenges of insufficient different domains for the model to learn, domain augmentation is straightforward and multiple methods have been proposed to generate novel domains and images through mixup (Yan et al., 2020), mixing of statistics (Zhou et al., 2021), uncertainty modeling (Li et al., 2022b; Zhou & Konukoglu, 2023) and convex combination (Albuquerque et al., 2019). However, these methods generally interpolate the existing training domains to generate novel domains that still fall within the convex hall of available domains (Albuquerque et al., 2019). Consequently, the constrained number of source domains hampers the expressiveness of these methods,

continuing to act as a performance bottleneck. Humans harness the innate ability of the human brain to create novel domains as illustrated in (Shu et al., 2023; Radford et al., 2021) where a pre-defined set of novel domains and styles are utilized. However, this also requires human labor which fails to scale to larger sizes.

On the other hand, Large language models (LLMs) (Brown et al., 2020) have been shown to encapsulate a vast wealth of knowledge and simulate human cognitive processes. Thus, a pertinent question emerges: Can one harness the power of LLMs to produce novel domains and relevant knowledge, thereby replacing the human in the above training process? Stemming from this primary query, we investigate how we can extract knowledge of a specific task and produce novel domains from LLMs. A subsequent research question is: How can we leverage this text-centric knowledge from LLMs to instruct an image system that processes pixel input? State-of-the-art text-to-image generation models such as Imagen (Saharia et al., 2022), Stable Diffusion (Rombach et al., 2022b) and GLIDE Nichol et al. (2021) exhibit a great capability to synthesize photo-realistic images positioning them as the optimal conduit between textual and visual realms. Finally, we seek to answer to what extent the synthesized images based on knowledge can serve as good representation learners that can generalize to unseen testing domains. Following these problems, we are the first study to design a new paradigm that leverages the knowledge of LLMs to extrapolate novel domains for training better generalizable and sample-efficient models.

**Our findings.** In both single and multi-domain configurations, we demonstrate that synthetic data in the extrapolated novel domains markedly outperforms baseline results across various datasets. Data synthesized via the knowledge from LLMs excels compared to the synthetic data directly generated from text-to-image generation models. This underscores the ability of LLMs to effectively extrapolate novel domains and integrate prior knowledge into the model.

Secondly, we underscore the scalability of our approach by highlighting that as the number of domains escalates, the performance correspondingly improves. Intriguingly, this trend diverges from the outcomes observed when merely augmenting the number of images in synthetic data directly produced by text-to-image generation models reported in (Azizi et al., 2023; He et al., 2022). This further demonstrates the pivotal role of the knowledge derived from LLMs in mitigating overfitting to synthetic data.

The above empirical evidence has illustrated the promise of leveraging the knowledge from the LLMs and a realization that can bridge the knowledge for learning arbitrary tasks. We further propose a more challenging setting termed, data-free domain generalization, that endeavors to generalize to unseen testing domains with synthetic data only. This obviates the necessity for training data collection, thereby offering potential time and financial savings. Remarkably, our proposed method exhibits near-supervised performance in this setting, even surpassing the supervised baseline by approximately 1-2% on VLCS.

## 2 METHOD

We motivate our method from the perspective of the theoretical error bound for domain generalization. We first provide the notation for the theoretical framework. Then we motivate our research problem from the domain generalization error bound, i.e. limited number of source domains, which leads to a larger error bound. Then we propose a proxy method that approximates the meta-distribution with a proxy distribution. We give a new error bound on this method. Lastly, we propose one realization of our method by using LLMs to approximate the meta-distribution and text-to-image generation models to bridge the text-centric knowledge with the input pixel space.

### 2.1 THEORETICAL BOUND

**Notation.** Let $\mathcal{X}$ denote the observation space and $\mathcal{Y} = \{1, -1\}$ the output space. Denote $P_{XY}$ as the joint probability of the joint space of $\mathcal{X} \times \mathcal{Y}$ and assume a meta distribution $\mu$ and n domains $P_{XY}^{(1)}, \cdots, P_{XY}^{(i)}, P_{XY}^{(n)}$ are i.i.d realizations from $\mu$. A decision function is a function $f \in \mathcal{F} : \mathcal{X} \to \mathcal{Y}$ predicts $\hat{y}_i = f(x_i)$. We denote $l : \mathcal{Y} \times \mathcal{Y} \to \mathbb{R}_+$ a loss function and define the generalization error of a decision function as

$$\mathcal{L}^\mu(f) = \mathbb{E}_{P_{XY} \sim \mu} \mathbb{E}_{(x,y) \sim P_{XY}} [l(f(x), y)] \tag{1}$$

Since we have no access to $\mu$ and all the realizations $P_{XY}^{(1)}, \cdots, P_{XY}^{(i)}, P_{XY}^{(n)}$ but sampled images from these realizations, we can derive an empirical error:

$$\hat{\mathcal{L}}^{\mu}(f) = \sum_{i=1}^{n} \sum_{j=1}^{m} l(f(x_i^{(j)}), y_i^{(j)}) \tag{2}$$

It's easy to see that when $n \to \infty, m \to \infty$, $\hat{\mathcal{L}}^{\mu}(f)$ converges to $\mathcal{L}^{\mu}(f)$, which gives the intuitive sense that increasing $m$ and $n$ gives us better-approximated solutions. Then we can derive the generalization bound with standard empirical Rademacher complexity bound Li et al. (2022a).

**Lemma 1** *For a 1-Lipschitz loss $l$, with confidence at least $1 - 2\delta$ and for all $f \in \mathcal{F}$, we have*

$$\mathcal{L}^{\mu}(f) \leq \hat{\mathcal{L}}^{\mu}(f) + 2\mathcal{R}_{mn}(\mathcal{F}) + 2\mathcal{R}_{n}(\mathcal{F}) + 3\sqrt{\frac{\ln(2/\delta)}{2mn}} + 3\sqrt{\frac{\ln(2/\delta)}{n}}$$

*where $\mathcal{R}(\mathcal{F})$ standard empirical Rademacher complexity on function class $\mathcal{F}$.*

Now we show that both the number of domains $n$ and the number of images observed from each domain $m$ is negatively correlated to the upper bound of generalization error. This motivates us to increase $n$ and $m$, which is difficult due to the inaccessible $\mu$ and $P_{XY}^{(1)}, \cdots, P_{XY}^{(i)}, P_{XY}^{(n)}$. Prior arts have proposed various methods to generate novel domains but the majority falls in the interpolation of existing domains, failing to effectively increase $n$. However, we can approximate $\mu$ by $\mu'$ sufficiently close to $\mu$ that can be sampled.

**Definition 1** *We define the distance between the two distributions as*

$$D(\mu, \mu') = \sup_{f \in \mathcal{F}} |\mathcal{L}^{\mu'}(f) - \mathcal{L}^{\mu}(f)|$$

With the following assumption,

**Assumption 1** *We assume the distance $D(\mu, \mu') \leq \epsilon$.*

we can derive a bound through the approximated $\mu'$.

**Theorem 1** *With confidence at least $1 - 2\delta$ and for all $f \in \mathcal{F}$, we have*

$$\mathcal{L}^{\mu}(f) \leq \hat{\mathcal{L}}^{\mu'}(f) + 2\mathcal{R}_{mn}(\mathcal{F}) + 2\mathcal{R}_{n}(\mathcal{F}) + 3\sqrt{\frac{\ln(2/\delta)}{2mn}} + 3\sqrt{\frac{\ln(2/\delta)}{n}} + \epsilon$$

By replacing $\mu$ with $\mu'$, we now have control over $\hat{\mathcal{L}}^{\mu'}(f)$, $m$ and $n$ as we can sample as many domains and images from $\mu'$ as possible. This is obtained at the cost of $\epsilon$, which we assume to be small.

**Remark 1** *We also note that as $n$ and $m$ increase, the upper bound of the generalization error decreases, which gives us better generalization errors.*

## 2.2 DOMAIN EXTRAPOLATION WITH LLMS

Given the aforementioned theoretical boundary, our objective is to approximate $\mu$ with $\mu'$. Humans, as evidenced in Shu et al. (2023); Radford et al. (2021), can be a good approximation. Nonetheless, human intervention is expensive and not scalable to larger datasets. Conversely, LLMs not only embody a vast expanse of knowledge (Petroni et al., 2019) and exhibit comparable reasoning capabilities (Qiao et al., 2023), but they also present the benefit of being amenable to extensive sampling. After sampling the domain distribution from meta distribution $\mu'$, we need to further sample from the domain distribution to generate images in particular novel domains. As discussed in Section 1, this provides a bridge from the text-based knowledge output by the LLMs and the input pixel space of vision systems. Text-to-image generation models (e.g. stable diffusion Rombach et al. (2022a)) exhibit the great capability to output photo-realistic images through inputting texts positioning them as the optimal bridge between textual and visual realms. The synthetic images of extrapolated novel

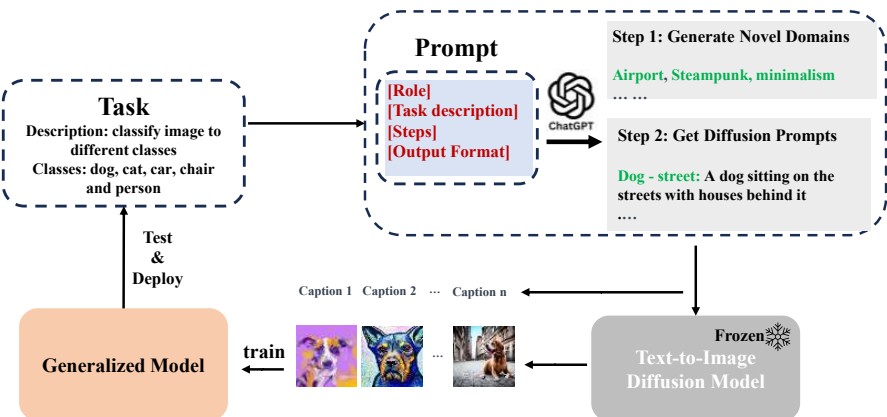

Figure 1: Overall pipeline of our paradigm: *Extrapolation of novel domains via the knowledge of LLMs*, a novel learning paradigm where knowledge from LLMs assists the training of generalizable models via text-to-image models in a completely data-free fashion.

domains are used to augment the original dataset or train the models solely in a data-free fashion. An overall illustration of our paradigm can be seen in Figure 1.

**Extracting Knowledge from LLMs.** The objective is to approximate $\mu$ via LLMs as close as possible. This introduces a constraint whereby the generated novel domains must reside within the high-density regions of distribution $\mu$. To ensure adherence to this criterion, we purposefully instruct the LLMs to conceive the most plausible and reasonable domains where a particular class would realistically exist. To better guide LLMs to understand the instruction and generate the required response accordingly, we craft system prompts that include role description ([Role]) and task description ([Task Description]), as illustrated by the example in Figure 2. Numerous strategies exist to solicit knowledge and novel domains from LLMs.

- Dataset-wise query. The most direct approach entails querying the LLMs with comprehensive dataset information (i.e. all of the class names) and instructing the model to produce $n$ novel domains. However, as the marginal distribution for each class might exhibit minimal overlap (worse when the number of classes grows), it becomes considerably intricate to sample novel domains that are both plausible and likely for all classes.

- Class-wise query. Thus, we propose to query the LLMs for novel domains of specific classes. For each class in the task, we query the LLMs for knowledge and $n$ novel domain information specific to that class. We repeat the process one class after another until all of the classes are iterated. We provide a example prompt in Figure xxx

**Bridging text and pixel with text-to-image generation models.** After obtaining a number of the most plausible and reasonable domains of a specific class, we transform the text-centric knowledge from LLMs to pixel space by text-to-image generation models. This process is exactly the realization of sampling $X$ from $P_X^{(i)}$ where $P_X^{(i)}$ is the $i$th domain generated by $\mu'$ (i.e. the LLM). Numerous strategies exist to prompt text-to-image generation models conditioned on class and domain information.

- Template prompt. The most immediate strategy involves employing templates as prompts (e.g., "an image of [CLASS_NAME] in the domain of [DOMAIN_NAME]"). However, the limitation lies in its lack of diversity: utilizing the identical prompt to produce multiple images results in images bearing resemblance to one another.

- LLM generated prompt. Thus, we propose to query the LLMs for prompts conditioned on the class name and domain information acquired in the previous step. As illustrated in Figure xxx, we craft system prompts that specifically tailor the LLM to generate prompts for text-to-image generation models and generate multiple prompts for each of the novel domains of each class.

**Filtering noisy images with CLIP.** Synthetic images are by their nature noisy since we have limited control over text-to-image generation. A few noisy examples are listed in Appendix C. Conse-

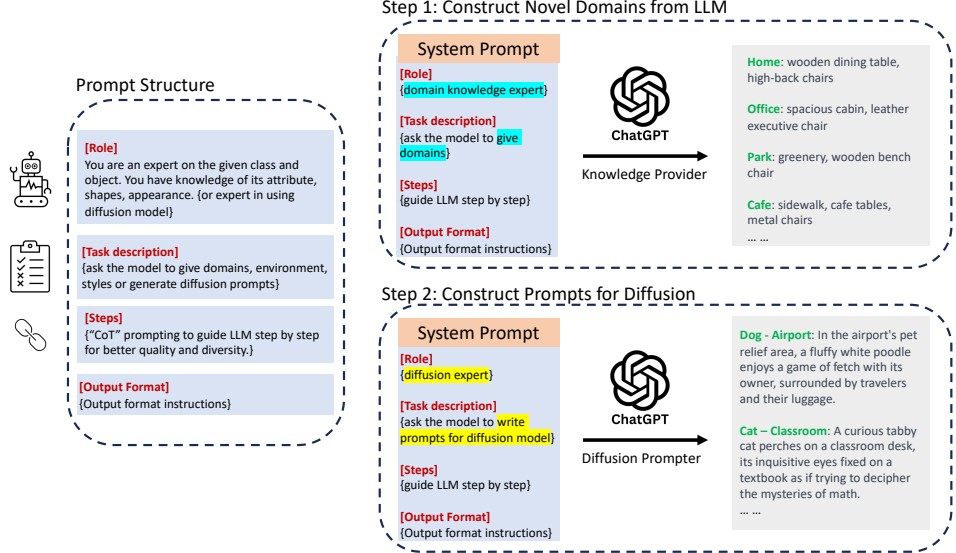

Figure 2: Knowledge extraction pipeline. We first employ various SOTA prompting methods: e.g. "Chain of Thought (Wei et al., 2022)" (CoT) prompting, role prompting to extract domains from LLM (Step 1) and automatically generate prompt for a Text-to-Image model. (Step 2)

quently, we apply a post-precessing step using CLIP Radford et al. (2021) to filter the images that contain no class of interest. After acquiring the synthetic data generated conditioned on extrapolated novel domains and classes, we augment the training sets with these high-fidelity, photo-realistic images and perform standard training procedures.

**Data-free Domain Generalization.** To illustrate and better underscore the argument that the knowledge from the LLMs and a realization that can bridge the text-driven knowledge and the pixel input space is sufficient to learn a generalized model for any task, we put forth data-free domain generalization that aims to learn a generalized model in the absence of any collected data. Under this setting, only synthetic data is used during training and all original training domains are used as testing domains. This setting not only serves as a more difficult benchmark for our method but also unveils the potential capability of generalizing to any task with only task information, the help from LLMs and a realization that connects text-based knowledge to pixel space.

## 3 EXPERIMENTS

The objective of our experiments is to (i) demonstrate that knowledge from LLMs successfully extrapolates novel domains and leads to performance benefits grounded by theoretical bounds. (ii) Investigate the most efficient and effective approach for extracting knowledge and sampling from text-to-image models. (iii) Analyze to what extent the synthetic images generated condition on LLMs' knowledge can serve as good representation learners which can train models that generalize to unseen testing domains.

### 3.1 SETUP

**Datasets.** We evaluate generalization to domain shift using four multi-domain datasets in DomainBed Gulrajani & Lopez-Paz (2020), namely PACS, VLCS, OfficeHome and DomainNet. We follow the train-validate-test split of each dataset as in Gulrajani & Lopez-Paz (2020) and use the training-domain validation set to perform the hyperparameter search.

**Evaluation.** To comprehensively evaluate our method, We experiment on both the leave-one-out evaluation protocol and single-domain generalization protocol. In addition, we propose the data-free domain generalization to evaluate whether it is possible to train a generalizable model in a data-free fashion with only task information, the knowledge from LLMs and text-to-image models that bridge the text space to pixel space.

Table 1: Main results on DomainBed Benchmark. We adopt multi-domain leave-one-out evaluation, single domain generalization and data-free generalization to evaluate our methods. CLIP adopts ViT-B16 as the backbone.

| Algorithm | VLCS | PACS | OfficeHome | DomainNet | Avg |
|---|---|---|---|---|---|
| *Leave-one-out Evaluation* | | | | | |
| Mixup | $78.1 \pm 0.3$ | $86.8 \pm 0.3$ | $68.0 \pm 0.2$ | $39.6 \pm 0.1$ | 68.1 |
| MMD | $77.9 \pm 0.1$ | $87.2 \pm 0.1$ | $66.2 \pm 0.3$ | $23.5 \pm 9.4$ | 63.7 |
| RSC | $77.8 \pm 0.6$ | $86.2 \pm 0.5$ | $66.5 \pm 0.6$ | $38.9 \pm 0.6$ | 67.4 |
| VREx | $78.1 \pm 0.2$ | $87.2 \pm 0.6$ | $65.7 \pm 0.3$ | $30.1 \pm 3.7$ | 65.3 |
| IRM | $76.9 \pm 0.6$ | $84.5 \pm 1.1$ | $63.0 \pm 2.7$ | $28.0 \pm 5.1$ | 63.1 |
| SWAD | $79.1 \pm 0.4$ | $88.1 \pm 0.4$ | $70.6 \pm 0.3$ | $46.5 \pm 0.2$ | 66.9 |
| MIRO | $79.0 \pm 0.2$ | $85.4 \pm 0.4$ | $70.5 \pm 0.4$ | $44.3 \pm 0.2$ | 65.9 |
| ERM | $77.2 \pm 1.0$ | $84.4 \pm 0.8$ | $64.8 \pm 0.4$ | $43.6 \pm 0.1$ | 67.5 |
| + ours | $78.5 \pm 0.4$ | $88.0 \pm 0.3$ | $70.0 \pm 0.1$ | $45.2 \pm 0.1$ | 70.4 |
| $\Delta$ | +1.3 | +3.6 | +5.2 | +1.6 | +2.9 |
| ERM + EMA | $78.8 \pm 0.6$ | $87.8 \pm 0.3$ | $70.5 \pm 0.1$ | $46.0 \pm 0.1$ | 70.8 |
| + ours | $80.2 \pm 0.3$ | $90.3 \pm 0.4$ | $74.6 \pm 0.2$ | $47.5 \pm 0.3$ | 73.2 |
| $\Delta$ | +1.4 | +2.5 | +4.1 | +1.5 | +1.4 |
| CLIP Zero-shot | 80.1 | 96.2 | 83.0 | 58.5 | 79.5 |
| CLIP Finetune | $82.4 \pm 0.1$ | $95.3 \pm 0.2$ | $84.8 \pm 0.1$ | $59.9 \pm 0.1$ | 80.6 |
| + ours | $82.4 \pm 0.4$ | $96.3 \pm 0.1$ | $86.5 \pm 0.2$ | $62.5 \pm 0.4$ | 81.9 |
| $\Delta$ | +0.0 | +1.0 | +1.7 | +2.6 | +1.3 |
| *Single Domain Generalization* | | | | | |
| ASA | - | 67.0 | - | - | - |
| Pro-RandConv | - | 67.0 | - | - | - |
| CPerb | - | 73.3 | - | - | - |
| RSC | $55.9 \pm 0.6$ | $58.4 \pm 3.5$ | $41.4 \pm 0.9$ | - | 51.9 |
| ERM | $55.7 \pm 1.7$ | $61.7 \pm 1.1$ | $49.2 \pm 1.8$ | - | 55.5 |
| + ours | $76.3 \pm 0.2$ | $83.9 \pm 0.9$ | $64.7 \pm 0.2$ | - | 75.0 |
| $\Delta$ | +20.6 | +22.2 | +15.5 | - | +19.4 |
| ERM+EMA | $64.9 \pm 1.7$ | $65.9 \pm 0.3$ | $57.4 \pm 0.7$ | - | 62.7 |
| + ours | $78.0 \pm 0.1$ | $87.6 \pm 0.6$ | $69.4 \pm 0.3$ | - | 78.3 |
| $\Delta$ | +13.1 | +21.7 | +12.0 | - | +15.6 |
| *Data-free Generalization* | | | | | |
| ERM + ours | $73.9 \pm 0.3$ | $82.5 \pm 0.9$ | $61.9 \pm 0.1$ | - | 72.8 |
| ERM + EMA + ours | $79.9 \pm 0.6$ | $86.9 \pm 0.1$ | $67.4 \pm 0.2$ | - | 78.1 |

**Baseline.** We set two baselines for our experiments, namely empirical risk minimization (ERM) and ERM with exponential moving average (ERM + EMA) which is demonstrated to be more stable and effective than ERM Arpit et al. (2022). We adopt ERM + EMA to perform ablation study and analysis since it's performance is more stable and more correlated to the validation accuracy Arpit et al. (2022).

**Implementation.** All experiments use ResNet50 pretrained on ImageNet1k as the image encoder unless otherwise stated. We remove the dropout and follow the rest of the implementation as in Gulrajani & Lopez-Paz (2020) since dropout is reported to have a negative impact on some of the DG methods Huang et al. (2022), e.g. RSC Huang et al. (2020). We adopt GPT-4 to extract novel domain knowledge and leverage Stable Diffusion 2 Rombach et al. (2021) as the text-to-image generation model. We use one A100 GPU to generate synthetic images. All experiments of training ResNet50 and CLIP ViT-B16 model can be run on 1 RTX3090 GPU.

## 3.2 MAIN RESULTS

We perform two existing evaluations on the four datasets in DomainBed benchmarks. Additionally, we propose a more challenging evaluation to further investigate to the synthetic images generated condition on LLMs' knowledge can serve as good representation learners.

**Leave-one-out evaluation.** Leave-one-out Evaluation leaves one domain as the testing domain and uses the rest as training domains. For our method, all of the synthetic images are used as an

additional domain as an augmentation to the source domains. As per Table 1, augmenting with the novel domain synthetic images leads to a consistent improvement (as large as 5.2%) over the ERM and ERM + EMA baselines. On average, we achieve a 2.9% and 2.38% improvement over ERM and ERM + EMA baselines respectively. Our method also achieved a significant improvement (1.33% on average) over the CLIP fine-tuned baseline

**Single Domain Generalization.** Single domain generalization Evaluation leverages a single domain for training purposes and subsequently assesses the outcomes on the remaining domains. This scenario presents a greater challenge when juxtaposed with the Leave-one-out setting due to the model's exclusive exposure to just one domain during its training phase. Such a setting accentuates the issue of restricted availability of source domains. Considering our methodology does not impose assumptions on either the source domains or the model, but instead extrapolates novel domains via LLMs to augment the training set, it is optimally suited for this specific context. Empirical evidence underscores its exceptional efficacy and with merely one source domain of real images, our results closely mirror, and at times even surpass, those obtained in a multi-domain configuration. Specifically, we achieve the highest of 78.0%, 87.6%, 69.4% on the three datasets, outperforming the ERM baseline with multiple source domains by margins of 0.8%, 3.2% and 4.6% respectively. Compared to baselines, our method achieves a remarkable improvement of over 10% across all datasets and baselines. This evidences that our methodology substantially mitigates the challenges associated with restricted source domains, rendering it particularly optimal and effective in scenarios where source domains are unavailable, such as single domain generalization. **Data-free Domain Gener-**

Table 2: Comparison with baseline and SOTA augmentation-based DG methods with multi-domain leave-one-out evaluation on PACS. Experiments are done on PACS dataset.

| Algorithm | A | C | P | S | Avg |
|---|---|---|---|---|---|
| DSU | $84.0 \pm 2.0$ | $80.4 \pm 1.1$ | $96.1 \pm 0.5$ | $\mathbf{81.6 \pm 1.3}$ | 85.5 |
| ERM | $87.1 \pm 0.8$ | $77.8 \pm 0.9$ | $96.1 \pm 1.0$ | $76.7 \pm 0.8$ | 84.4 |
| +AutoAug | $\underline{88.0 \pm 0.5}$ | $78.5 \pm 0.7$ | $96.3 \pm 0.0$ | $79.1 \pm 1.0$ | 85.5 |
| +RandAug | $86.7 \pm 0.5$ | $78.3 \pm 2.0$ | $97.3 \pm 0.4$ | $\underline{80.1 \pm 0.8}$ | 85.6 |
| +larger batch-size | $85.9 \pm 2.5$ | $\underline{80.9 \pm 0.4}$ | $\underline{97.5 \pm 0.2}$ | $78.2 \pm 1.6$ | 85.6 |
| +sythetic (class template) | $87.7 \pm 0.6$ | $80.4 \pm 0.3$ | $96.9 \pm 0.3$ | $74.7 \pm 1.3$ | $\underline{85.9}$ |
| +ours | $\mathbf{91.7 \pm 0.7}$ | $\mathbf{82.4 \pm 1.0}$ | $\mathbf{97.9 \pm 0.0}$ | $80.0 \pm 1.4$ | $\mathbf{88.0}$ |

**alization.** To further push the limits of our method, we propose an even harder evaluation setting, data-free domain generalization, where only knowledge of task, i.e. the classes and definition of each class is available and no available data of any kind. We directly train models on the synthetic images generated conditioned on novel domain knowledge. Then the model is tested on all the available real image domains for evaluation. Given the noisy and unstable nature of synthetic images, our method surprisingly achieves remarkable results achieving near-supervised performance. Specifically, data-free ERM+EMA gives the highest performance of 79.9%, 86.9%, 67.4% with only less than 1% gap between multi-domain and largely surpasses single-domain baselines. Notably, data-free ERM+EMA presents an accuracy of 79.9% on VLCS outperforming the multi-domain supervised baseline by more than 1%. With the knowledge injected and novel domain extrapolated, this empirical result illustrates the promise of achieving generalization in a completely data-free fashion free of laborious data collection and annotation.

**Comparison with augmentation-based DG methods.** We compared with SOTA augmentation methods in Table 2 including MixStyle, DSU, AutoAug and RandAug, where our method demonstrates an improvement of more than 2%.

## 3.3 ABLATION STUDY AND ANALYSIS

To fully understand the performance of our method, we perform an ablation study by first providing two baselines building upon ERM with minor modifications. First, we provide **larger batchsize** baseline, which is used to ablate the influence of larger batch sizes incurred by augmented data. Then, we provide **class prompt** baseline, which prompts the text-to-images generation model to generate synthetic images with the template "An image of [CLASS]". This ablates the influence brought by text-to-image models and further underscores the importance of LLMs' knowledge regarding the novel domain.

**Effectiveness of filtering.** As mentioned in Section 2, we introduce CLIP model to perform an additional filtering to discard the noisy images generated by the text-to-image generation model. Results can be seen in (a) of Figure 3.3 where filtering leads to around 1% improvement.

**Comparison between different knowledge extraction.** We provide three approaches to extract knowledge regarding the novel domains of particular classes. Comparison can be seen in (b) of Figure 3.3, where we show that, in overall, class-wise combined with LLM-generated prompt leads to better performance than class-wise query only and data-wise query. This is because class-wise query provides more plausible and reasonable novel domains given some class and LLM-generated prompt further extracts knowledge regarding this novel domain and increases diversity in generation.

**Scaling.** It has been widely reported that data generated by generation models negatively impacts the model, especially when the number of synthetic images grows at scale He et al. (2022); Azizi et al. (2023). To this end, we investigate whether the performance increases scales with more synthetic data. There are two dimensions we can scale up our synthetic images of novel domains, i.e. domain dimension and sample dimension. We first perform a scaling on domain dimension in Table 3.3. Results demonstrated that our method scales to larger sizes of synthetic data and performance keeps growing without saturation. This is in stark contrast to synthetic images generated by class templates, where performance peaks at 4096 images per class and decreases, which is consistent with previous studies He et al. (2022); Azizi et al. (2023). Our method, on the other scales to larger synthetic sizes, which we believe is due to the new knowledge injected by LLMs that benefits generalization. We also test the scaling ability on the sample dimension and show in Table 3 that our method improves scaling and reduce overfitting to synthetic data.

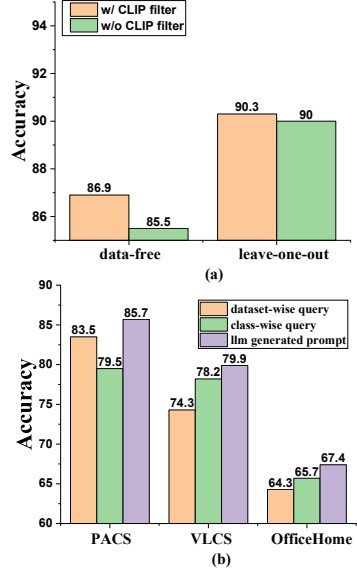

| #domain | #image | Novel domain | class template |
|---|---|---|---|
| 16 | 1024 | $76.4 \pm 1.0$ | $58.9 \pm 0.9$ |
| 32 | 2048 | $83.2 \pm 0.1$ | $59.0 \pm 0.2$ |
| 48 | 3072 | $84.0 \pm 0.8$ | $59.4 \pm 0.2$ |
| 64 | 4096 | $84.5 \pm 0.5$ | $\mathbf{60.9 \pm 0.2}$ |
| 80 | 5120 | $84.7 \pm 0.2$ | $58.6 \pm 1.0$ |
| 96 | 6144 | $86.6 \pm 0.3$ | $58.8 \pm 1.0$ |
| 112 | 7168 | $\mathbf{86.9 \pm 0.1}$ | $57.5 \pm 1.1$ |

Table 3: Scaling the training dataset by adding more novel domains. Each novel domain consists of 64 images. To facilitate fair comparison, we scale the class template method by the same amount of images.

| #domains | #images | Novel domain |
|---|---|---|
| | 32 | $85.4 \pm 0.0$ |
| | 64 | $85.1 \pm 0.5$ |
| 64 | 128 | $86.1 \pm 0.3$ |
| | 160 | $\mathbf{87.3 \pm 0.5}$ |
| | 256 | $87.1 \pm 0.3$ |

Table 4: Scaling the training dataset by adding more images while fixing the number of novel domains to 64. #domain and #image are measured per class.

Figure 3: (a) Effectiveness of CLIP filtering. (b) Comparison between different knowledge extraction methods.

**Visualization.** We provide visualization of generated images from three novel domains of PACS dataset in Figure 4 (last four columns). We compare them to the real images in PACS (first two columns). We can see that the generated novel domains are by no means an interpolation of the real domains and in fact varies from the real domains by a large margin. We further illustrate that our method takes one step further towards "truly" extrapolation of novel domains without human labor. We provide more visualization in Appendix.

## 4 RELATED WORK

**Domain Generalization.** Various approaches have been proposed to solve this problem, such as domain alignment Li et al. (2018b;c), meta-learning Li et al. (2018a); Balaji et al. (2018), ensemble learning Cha et al. (2021); Arpit et al. (2022) and augmentation-based Zhou & Konukoglu (2023); Zhou et al. (2021); Li et al. (2022b); Xu et al. (2020); Zhou et al. (2020); Albuquerque et al. (2019).

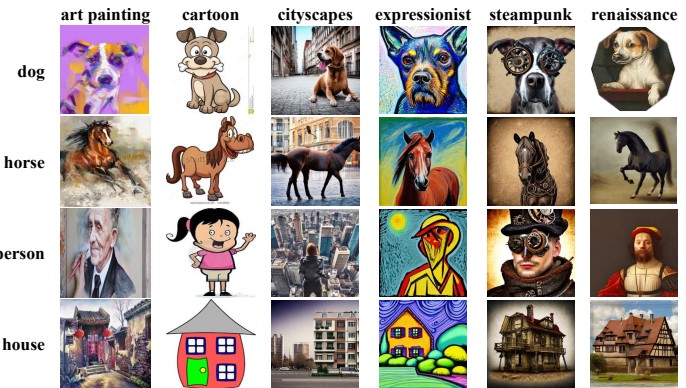

Figure 4: Examples of synthetic images conditioned on novel domain knowledge from LLM. The first two columns (i.e. art painting and cartoon) are selected from PACS datasets while the rest four columns are images generated based on the novel domains (i.e. cityscapes, etc) provided by LLMs.

Augmentation-based methods are closely related to this work, both with the intention of generating more source domains to approximate the expected generalization error. However, these methods resort to interpolation of existing domains and fail to extrapolate the "truly" novel domains. For instance, MixStyle Zhou et al. (2021) mixes the statistics of two samples by linear interpolation. More recently, with the advent of vision-language models such as CLIP Radford et al. (2021) and Stable Diffusion Rombach et al. (2021), researchers propose to utilize Stable Diffusion to identify and cure shortcuts Wu et al. (2023) or CLIP to generate novel domain augmentation Vidit et al. (2023). However, they all require some form of human labor to pre-define a set of domains or styles, which makes them laborious and not scalable. Our work aims to solve this problem and achieve genuine domain extrapolation.

**Language scaffolded vision** aims to develop better and more robust vision systems with the help of language. Our method also falls within this category. Clipood Shu et al. (2023) proposes to fine-tune a CLIP model to adapt the downstream DG tasks by a text similarity aware loss. Min et al. (2022) utilize an RNN as an explanation network enforcing the model to self-explain, thereby increasing the robustness. Yang et al. (2023) utilize language models to produce a comprehensive set of bottleneck features and leverage CLIP to classify. With the help from LLMs, Yang et al. (2023) has pushed the performance of the bottleneck network to SOTA. Despite many works proposed, this research, to the best of our knowledge, is the first endeavor to investigate the potential of a Large Language Model (LLM) in facilitating the training of a robust and generalizable vision model.

**Large Language Models.** Recent advances in NLP, as evidenced by (Brown et al., 2020; Ouyang et al., 2022)) highlight the impressive capabilities of Large Language Models like Chat-GPT, GPT4 (Brown et al., 2020), and Llama 2 (Touvron et al., 2023). These models glean diverse knowledge from vast training data sourced from the Internet, positioning LLMs as next-generation knowledge bases for various tasks. Motivated by studies showcasing the vast knowledge (Alivanistos et al., 2022; Petroni et al., 2019) and the exceptional reasoning ability (Huang & Chang, 2023; Qiao et al., 2023; Wei et al., 2022) within LLMs, we aim to harness this knowledge for the training of robust vision models.

## 5 CONCLUSION

The limited availability of domains has been a prevailing problem in Domain Generalization. In this work, we propose the first data-free learning paradigm that leverage the knowledge and reasoning of LLMs to extrapolate novel domains. By bridging the text-centric knowledge and pixel input space by sampling from text-to-image generation models, we are able to train generalizable models with task information only. The synthetic images can be used to augment the existing dataset or train a model in a data-free fashion. Extensive experiments have demonstrated that our method achieves significant improvements over baselines and the state-of-the-art by a significant margin. We also demonstrate a promising learning paradigm where LLMs' knowledge combined with text-to-image generation models are sufficient to train a generalizable model to any task.

**Limitations** is in Appendix D.

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
