

Figure 5: Examples of synthetic images conditioned on novel domain knowledge from LLM. The first two columns (i.e. Caltech101 and VOC2017) are selected from VLCS datasets while the rest three columns are images generated based on the novel domains (i.e. fairytale, etc) provided by LLMs

## A  PROOF OF THEOREM 1

With confidence at least $1 - 2\delta$ and for all $f \in \mathcal{F}$, we have

$$\mathcal{L}^{\mu}(f) - \hat{\mathcal{L}}^{\mu'}(f) = \mathcal{L}^{\mu}(f) - \mathcal{L}^{\mu'}(f) + \mathcal{L}^{\mu'}(f) - \hat{\mathcal{L}}^{\mu'}(f) \tag{3}$$

With Lemma 1, we have

$$\mathcal{L}^{\mu}(f) - \mathcal{L}^{\mu'}(f) + \mathcal{L}^{\mu'}(f) - \hat{\mathcal{L}}^{\mu'}(f) \tag{4}$$

$$\leq 2\mathcal{R}_{mn}(\mathcal{F}) + 2\mathcal{R}_n(\mathcal{F}) + 3\sqrt{\frac{\ln(2/\delta)}{2mn}} + 3\sqrt{\frac{\ln(2/\delta)}{n}} + \mathcal{L}^{\mu}(f) - \mathcal{L}^{\mu'} \tag{5}$$

$$\leq 2\mathcal{R}_{mn}(\mathcal{F}) + 2\mathcal{R}_n(\mathcal{F}) + 3\sqrt{\frac{\ln(2/\delta)}{2mn}} + 3\sqrt{\frac{\ln(2/\delta)}{n}} + \sup_f |\mathcal{L}^{\mu}(f) - \mathcal{L}^{\mu'}| \tag{6}$$

With the assumption that $D(\mu, \mu') = \sup_f |\mathcal{L}^{\mu}(f) - \mathcal{L}^{\mu'}| \leq \epsilon$, we have

$$\mathcal{L}^{\mu}(f) - \hat{\mathcal{L}}^{\mu'}(f) \tag{7}$$

$$\leq 2\mathcal{R}_{mn}(\mathcal{F}) + 2\mathcal{R}_n(\mathcal{F}) + 3\sqrt{\frac{\ln(2/\delta)}{2mn}} + 3\sqrt{\frac{\ln(2/\delta)}{n}} + \epsilon \tag{8}$$

which finishes the proof.

## B  VISUALIZATION

We provide more examples of synthetic images conditioned on novel domain knowledge from LLM. We present in Figure 5 the synthetic images of VLCS datasets.

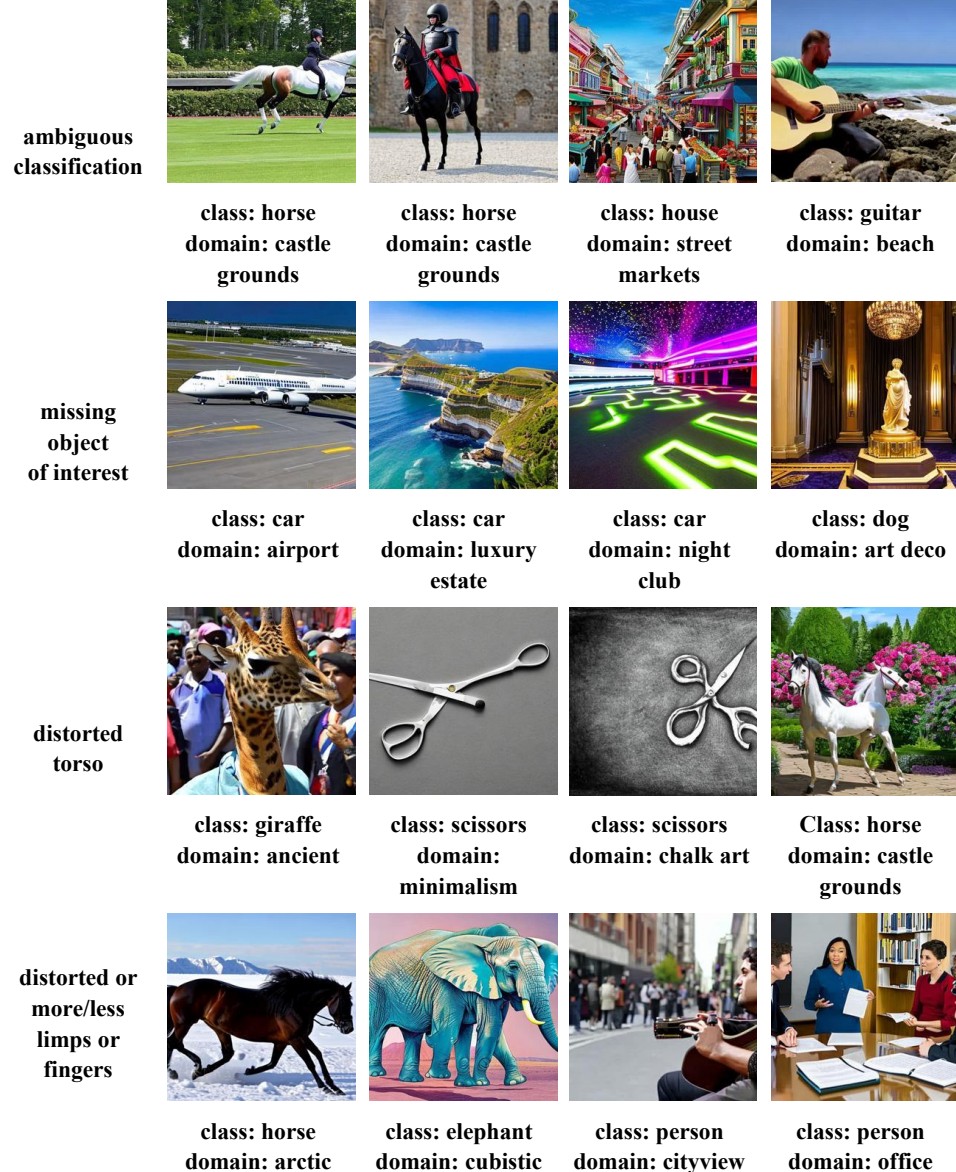

Figure 6: Examples of pitfalls of synthetic images.

## C    PITFALL OF TEXT-TO-IMAGE GENERATION MODELS

Text-to-image generation models are by nature noisy as no strict control can be achieved. We present some pitfalls (commonly reported by the community) that will insert noise and influence the training of a generalizable model. We show in Figure 6 where each row is a type of problem and below each image is the corresponding class and domain.

## D    LIMITATIONS

We utilize foundational models, including Large Language Models (LLMs) and text-to-image generators, both of which carry inherent biases. Vision models trained with these can thus inherit these biases. Our method struggles with the long-tail distribution challenge as these models favor common object representations over rarer ones. This bias exacerbates the long-tail issue, with common entities dominating generation. Moreover, current text-to-image models are domain-specific, performing well with natural images but faltering in areas like medical imaging.