# OpenReview forum: "Novel Domain Extrapolation with Large Language Models"
_ICLR.cc/2024/Conference — ICLR 2024 Conference Withdrawn Submission_

### Official Review · Reviewer_iNq6 · 2023-10-22

**Soundness:** 2 fair
**Presentation:** 4 excellent
**Contribution:** 2 fair
**Rating:** 3
**Confidence:** 5

**Summary:**

The paper aims to enhance Domain Generalization (DG) performance for visual classification tasks. It introduces an approach of exploiting the vast knowledge embedded in Large Language Models (LLMs) to generate entirely new domains, by employing a diffusion model to convert language into images. The proposed method is evaluated on the PACS, VLCS, OfficeHome, and DomainNet datasets and compared with other state-of-the-art DG techniques.

**Strengths:**

1. The paper is generally well-structured and well-written;

2. The exploration of Large Language Models (LLMs) and Diffusion models to address the Domain Generalization problem in visual tasks is a valuable endeavor, given their strong capabilities.

**Weaknesses:**

1. The proposed methodology is not that novel. The technique primarily employs a Large Language Model (LLM) to generate sentences describing potential locations of the target class, and then applies a diffusion model to create training image samples. Essentially, this approach relies on the LLM and diffusion model for generating more training data.

2. The significant improvement compared to other single-source Domain Generalization (DG) methods is not surprising. As depicted in Figure 4, the paper directly generates images in a variety of styles to train models for PACS. Comparing models trained with such data to models with the original training data does not provide a fair comparison.

3. It would be beneficial for the authors to explore the application of their method on more challenging and more realistic DG datasets where current DG methods struggle, such as the FMoW-Wilds dataset (https://wilds.stanford.edu/datasets/#fmow).

4. Typo: “Figure xxx” in page 4.

**Questions:**

See weaknesses.

**Details Of Ethics Concerns:**

Not applicable.

---

### Official Review · Reviewer_8hCj · 2023-10-27

**Soundness:** 2 fair
**Presentation:** 3 good
**Contribution:** 1 poor
**Rating:** 3
**Confidence:** 5

**Summary:**

This paper addresses the problem of generating novel domains to enhance domain generalization. Specifically, it employs a large language model (LLM) to craft prompts that instruct text-to-image generation models to produce samples within new domains. Through experiments on datasets such as VLCS, PACS, and OfficeHome, it outperforms some domain generalization benchmarks and diffusion models that utilize class prompts. Additionally, it presents the results of a model trained solely on synthetic data—termed "data-free domain generalization"—and highlight its ability to surpass the supervised approaches.

**Strengths:**

1. This paper first use LLM to facilitate new domain generation, which is new to me.
2. The paper is easy to follow.

**Weaknesses:**

1. Lack of novelty. Their approach, which employs LLM to guide text-to-image generation, appears more as straightforward engineering than a revelation of deeper insights.

2. Loose connection between their theory and their method. While their theory suggests that increasing the number of novel domains would improve domain generalization performance, it doesn't persuasively argue why domain interpolation might be inferior to domain "extrapolation". Nor does it reason why we should combine LLM with text-to-image generation models to generate new domains. It feels as though the authors are attempting to retrofit a theory onto their method.

3. Overclaim. For example, the authors assert that "as the number of domains escalates, the performance correspondingly improves", citing this as a major contribution. However, according to Table 3, they've only tested up to 112 domains. While this demonstrates superior scalability compared to class-prompt based models, it's a leap to conclude that such augmentation methods wouldn't eventually plateau. In real-world scenarios, there could be thousands of distinct domains to consider.

**Questions:**

1. Why isn't Table 4 referenced in the paper?
2. Some crucial baselines are missing references. For example, in Table 1, what do "VREx", "SWAD", and "MIRO" methods correspond to?
3. Above Table 2, it would be clearer if "Data-free Domain Generalization" began on a new line.
4. What are "Figure 3.3" (mentioned under "Effectiveness of filtering") and "Table 3.3" (referenced in "Scaling")? I can't locate them in the paper.

---

### Official Review · Reviewer_h1jS · 2023-11-01

**Soundness:** 3 good
**Presentation:** 3 good
**Contribution:** 2 fair
**Rating:** 3
**Confidence:** 4

**Summary:**

This paper proposes a new method for domain generalization by leveraging large language models (LLMs) to extrapolate novel domains and generate synthetic data. The authors first query LLMs to extract novel domains and domain-specific knowledge for a given classification task. Then, they use the text descriptions from the LLM to generate synthetic images via text-to-image models like Stable Diffusion. The method is evaluated on DomainBed benchmarks in a leave-one-out setting and also more challenging single-domain and zero-shot settings. The main results demonstrate consistent and significant gains over baselines by using the LLM-guided synthetic data augmentation.

**Strengths:**

1. The idea of harnessing the knowledge and reasoning ability of LLMs for domain generalization is simple, straightforward, and well-motivated. Most prior works focus on interpolating existing domains.
2. Extensive experiments in diverse settings like leave-one-out, single-domain, and zero-shot demonstrate the effectiveness of the proposed method.

**Weaknesses:**

1. The authors only evaluate the proposed method on DomainBed. It remains unclear whether the method can be extended to more challenging benchmarks such as WILDS.

2. How does the proposed method compare with other augmentation techniques like MixStyle?

**Questions:**

Please check out the weakness

---

### Official Review · Reviewer_ek11 · 2023-11-06

**Soundness:** 1 poor
**Presentation:** 2 fair
**Contribution:** 1 poor
**Rating:** 1
**Confidence:** 4

**Summary:**

This project proposes using ChatGPT to generate prompts for Stable Diffusion as a means for generating novel domains for Domain Generalization. ChatGPT is fed a structured input specifying a Role, Task, and Output Format and it returns prompts corresponding to a domain. These prompts are then fed to Stable Diffusion, which returns images. Finally, Images are filtered using CLIP. Images generated using this process are added to the training set and domain generalization is tested on standard datasets including PACS, VLCS, OfficeHome, and DomainNet. Finally, ablations are reported to quantify impact of different design decisions including filtering, different knowledge extraction pipelines, and scaling.

**Strengths:**

1. Problem is important: Generalization is an extremely important problem as we move towards broad AI applications, which are deployed around us. Despite many Domain Generalization benchmarks, and a myriad approaches, we are still far from a viable solution. Thus, work along these lines is very relevant for the community.

2. Paper easy to follow: the manuscript does a good job of explaining what was done, and the design decisions involved.

3. As an engineering outcome, the pipelines presented here are useful, and could be of great help in the industry.

**Weaknesses:**

1. Fatal Flaw with Train/Test overlap: Stable Diffusion was trained with billions of images scraped from the internet, including images scraped from google search. PACS, VLCS, OfficeHome benchmarks were also created using the same process. Is there any reason to believe the PACS test domains were not included in LAION-5B used to train Stable Diffusion? If not, basically, training dataset contains a version of the test datasets evaluated on. This would explain why performance goes up.


2. Mathematical formulation flawed + Not even used:

- The manuscript presents a section on Theoretical Bounds for domain generalization. However, it is mentioned that the distribution of different domains are i.i.d. realizations of a meta-distribution. This is a major flaw, as this would mean that different domains have the same data distribution---thus making them just different subsets of the same domain. For a more plausible mathematical framework for different domains, please refer to: https://arxiv.org/pdf/2103.02503.pdf.

- Under this construction, there is no difference between increasing m and n. Since they are from the same distribution, and samples from each domain are also IID, we can just sample more m from one domain, and have the same impact as adding n.

- This construction is the reason why the generalization error decreases with both m and n under this construction. In a typical case of out-of-distribution generalization, it has been well documented that increasing domains (m) i.e. increasing diversity leads to improved performance, while increasing n (# images) cap out in performance after some time: https://par.nsf.gov/servlets/purl/10346962, https://www.nature.com/articles/s41593-018-0310-2, https://www.nature.com/articles/s42256-021-00437-5.


- The lemmas and theoretical bounds are not relevant to the work conducted here: the main contribution here is in engineering a system to design prompts for ChatGPT, which can then be passed to stable diffusion. It is unclear how this section fits into the work. If it is just to motivate, I think it is fairly well accepted in the DG community that increasing domains leads to improvement in performance out of distribution. However, the reason for this is believed to be building of invariances. The work with Uniform bounds with Rademarcher averages is conducted under knowledge of the data distribution, i.e. in-distribution generalization. Which is starkly different from the DG problem, where test domains is assumed to be out of distributino.

3. Other experimental flaws:

- When more domains are added, how is it ensured that number of images don't increase? If not ensured, it's unclear if accuracy increases because of new domains, or more new images.

- How do we know that LLM based knowledge extraction followed by stable diffusion will not lead to a convex hull? Have LLMs or Stable Diffusion known to be able to extrapolate past their training distributions?

- How is novelty ensured? In generation of novel domains, it is important to ensure that there is no overlap. How was this measured? How was it ensured?

Overall: This work presents a very useful engineering tool which can be of great use to people in industry, but it is deeply flawed as a scientific investigation which makes this manuscript not fit for publication.

**Questions:**

My main concern is of the train/test overlap. Unless that is addressed, this investigation is not on firm footing.